# Cleaning the *Medicago* Microarray Database to Improve Gene Function Analysis

**DOI:** 10.3390/plants10061240

**Published:** 2021-06-18

**Authors:** Francesca Marzorati, Chu Wang, Giulio Pavesi, Luca Mizzi, Piero Morandini

**Affiliations:** 1Department of Environmental Science and Policy, University of Milan, Via Celoria 10, 20133 Milano, Italy; Francesca.marzorati1@unimi.it; 2Department of Biosciences, University of Milan, Via Celoria 26, 20133 Milano, Italy; cwang@aeb-group.com (C.W.); giulio.pavesi@unimi.it (G.P.); luca.mizzi@unimi.it (L.M.)

**Keywords:** *Medicago*, MtGEA, transcriptomics, functional genomics, microarray, R programming, correlation analysis

## Abstract

Transcriptomics studies have been facilitated by the development of microarray and RNA-Seq technologies, with thousands of expression datasets available for many species. However, the quality of data can be highly variable, making the combined analysis of different datasets difficult and unreliable. Most of the microarray data for *Medicago truncatula*, the barrel medic, have been stored and made publicly accessible on the web database *Medicago truncatula Gene Expression atlas* (MtGEA). The aim of this work is to ameliorate the quality of the MtGEA database through a general method based on logical and statistical relationships among parameters and conditions. The initial 716 columns available in the dataset were reduced to 607 by evaluating the quality of data through the sum of the expression levels over the entire transcriptome probes and Pearson correlation among hybridizations. The reduced dataset shows great improvements in the consistency of the data, with a reduction in both false positives and false negatives resulting from Pearson correlation and GO enrichment analysis among genes. The approach we used is of general validity and our intent is to extend the analysis to other plant microarray databases.

## 1. Introduction

“Omic” technologies have been developed to investigate cellular molecules on a massive scale and they are classified according to the object studied: genes for genomics, RNA for transcriptomics, proteins for proteomics and metabolites for metabolomics [1]. Currently, ionomics (studying ions composition) is also arising as a major -omic science [2]. In transcriptomics, gene expression data coming from different tissues, conditions and genotypes can be obtained through different strategies, for example, different types of microarrays or RNA sequencing (RNA-Seq). These approaches allow to assess the expression level of most or nearly all the genes in an organism, and each experiment may envisage tens, if not hundreds, of measurements of different samples [3]. In this way, thousands of measurements on a genome-wide level are available for many species.

Microarrays are useful tools to explore genotypes and their interaction with the corresponding phenotypes, but the data produced by this technology require, however, processing for correct interpretation [4,5,6,7]. Even if the “death” of microarrays was predicted already in 2008 [8], there is still a wealth of data to be explored, and many novel datasets still appear in literature. Most of the microarray data are available online, e.g., on the *Gene Expression Omnibus* (GEO, https://www.ncbi.nlm.nih.gov/geo/, accessed on 13 May 2021) [9,10] or on dedicated websites, such as *The Arabidopsis Information Resource* (TAIR) (https://www.arabidopsis.org/, accessed on 13 May 2021) and *Medicago truncatula Gene Expression atlas* (MtGEA) (https://mtgea.noble.org/v3/, accessed on 13 May 2021) [11,12,13]. *Affymetrix m*icroarray data are organized in datasets with the list of probeset codes (the *Affymetrix* identifiers for each set of probe sequences designed to measure a transcript) as the first column. Each further column of the dataset contains the expression values for a single hybridization of a sample (tissue or condition). Samples are usually characterized by two to three replicates. Together, all samples from the same publication are referred to as an ‘experiment’.

*Medicago truncatula,* the barrel medic, is a small Mediterranean annual plant of the Fabaceae family, cultivated as a forage crop but extensively used as model organism for legumes. It is an autogamous plant, characterized by a short life cycle and a reduced genome size, allowing easy manipulation to study legume secondary metabolism [14,15]. Moreover, as many other legumes, *Medicago truncatula* establishes symbiotic relationships with nitrogen-fixing microorganisms; thus, it is used as model system to study this symbiosis [16,17,18].

In recent years, different transcriptomics resources have been developed for legumes, such as LegumeIP and LegumeGRN [19,20]. MtGEA is the gene expression atlas created specifically for *Medicago* spp., collecting most of the expression data obtained, using the *Affymetrix* GeneChip microarray technology [12,13,20,21]. In MtGEA, it is possible to explore the expression data for a gene of interest, which can be identified through its sequence, annotations or different identifiers, such as the *Affymetrix* probeset identifier, GO and KEGG annotation terms, gene name, and functional descriptions in natural language. Once identified, it is possible to perform different analyses on a gene or a gene list, such as the analysis of the expression profiles, co-expression studies, identification of genes showing differential expression among samples or experiments. Users can also download data in formats that are compatible with many analysis and visualization tools. The database is updated on a regular basis in order to include recent expression studies and updates on genome annotation [12,13].

In February 2020, we downloaded all the experimental data of *Medicago truncatula* collected in MtGEA, corresponding to 716 columns. Here, we report the analysis and cleaning of this dataset. After the cleaning, we performed a Pearson correlation and a Gene Ontology (GO) enrichment analyses on selected genes, working both on the original and cleaned datasets. The Pearson correlation coefficient assesses the strength of the linear relationship between two variables, expressed through a value between −1 and 1: a high correlation value means that two variables are strongly related, a negative value means that the variables are inversely related, whereas a small value means that the two variables are weakly associated [22,23]. Transcript correlation analysis is an important method to identify or confirm candidate genes involved in a pathway or process, as previously reported [24,25,26,27,28]. The correlation can be computed on the expression values as such or after Log-transformation, the latter being instrumental in revealing correlations holding also at low expression values.

The enrichment analysis is a procedure to interpret gene expression data identifying genes that are overrepresented in a large, provided set. We performed a GO term enrichment analysis, i.e., we identified GO categories overrepresented in selected gene sets [29,30]. By comparing the results of both analyses on the original and the cleaned datasets, we show significant changes in the lists of top correlators of several genes and the respective GO categories overrepresented in each list. Gene function predictions drawn from such lists may be substantially different, implying that the cleaning eliminates both false positive and negative correlators for a number of genes. We demonstrated that a proper cleaning of microarray datasets is required to find significant relations and GO functional enrichments among certain *Medicago* genes, results that are sustained by the literature and experimental evidence. We believe that the strategy developed is of general validity for the cleaning of expression microarray databases.

## 2. Results

### 2.1. Data

We downloaded all the *Affymetrix* microarray data from the *Medicago truncatula Gene Expression atlas* (MtGEA) (https://mtgea.noble.org/v3/, accessed on 13 May 2021) [12,13]. In February 2020, the complete dataset included 716 columns for 50,900 genes (Appendix A), whereas the dataset containing the means of the replicates, when available, of each sample comprised 274 values (Appendix A). As a start, we performed a Pearson correlation analysis, using the dataset of the means. Figure 1 shows a scatterplot for two probes, both referring to the same putative mevalonate kinase whose *Affymetrix* probe identifiers are Mtr.41545.1.S1_at and Mtr.16327.1.S1_at.

Each point of the scatterplot represents either the mean value of biological replicates (two or three) of the same sample (tissue or condition), or, in few cases, one single measurement. Strikingly, three values are localized in the space far from all the others. We decided to further investigate these outliers, first of all, by identifying the corresponding hybridizations, which belong to a group of three related samples (each with two replicates) obtained by laser capture microdissection (LCM): RT_LCM_arbuscular, RT_LCM_cortical and RT_LCM_adjacent [31].

### 2.2. Sum of the Expression Values

To understand if these three experiments present some peculiarity and thus could generate outliers for other genes, we computed the sum of the expression values of all genes for each hybridization in the dataset as a first index to check the quality of data (Appendix A). The mean of the sum of the expression values of the downloaded dataset is 1.81 × 10^7^, even if, in most of the hybridizations, the sum is around 2.0 × 10^7^. Figure 2 and Appendix A A focus on the group of samples RT_LCM [31], comparing the sum of the expression values to those of neighboring samples in the original dataset. This group of samples shows a “valley” in the sum, compared to most of the others.

Analyzing the results of the sum, we noticed other samples showing lower values compared to most of the others. Graphics for these experiments are presented in Appendix A. In particular, there are two additional groups of experiments with an extremely low sum, in the order of 10^5^. The two groups refer to specific studies, [32,33], respectively. Appendix A reports the sum of the expression values for these last two groups.

### 2.3. Pearson Correlation Coefficient

To further investigate the quality of data in the MtGEA database, we decided to compute the Pearson correlation coefficient for every pair of replicates in a sample in the downloaded dataset (Appendix A). Therefore, we did not consider further those samples with one single hybridization (30 hybridizations in total, listed in Appendix A). We focused on samples with at least two replicates, and we have considered acceptable a Pearson correlation coefficient among sample replicate pairs above a threshold value of 0.9. Replicate pairs with correlation values below this threshold are shown in Appendix A. Intriguingly, the Pearson correlation coefficients of the last two samples (row) in Appendix A show values that are identical to the sixth decimal. They refer to 24 replicates in total. Each of these replicates has two identities within each row or column in the correlation table (see Appendix A), suggesting that the data were duplicated (Appendix A). Indeed, expression values for 12 columns are duplicated (e.g., Root_A17_control_1 is identical to HairyRoot_WT_Myc_CK_1, and so on). These two groups refer to different studies [34,35], both characterized by 12 samples (GSE34155 and GSE34617). In addition, we also noted that there are samples (each with three replicates) whose names were duplicated. These refer to one study [12]: Nod_10dpi_1, Nod_10dpi_2, Nod_10dpi_3; Nod_14dpi_1, Nod14dpi_2, Nod14dpi_3 (Appendix A). In the dataset downloadable from MtGEA (Appendix A) there are 12 columns for these data, but only six can refer to original expression values. This leads to a shift in the attribution of names to samples, with data being referred to other ones starting from these columns.

### 2.4. Cleaning the Database

First, we removed the six duplicated names mentioned above and related to one study [12]; then, a total of 103 hybridizations were removed from the dataset, obtaining a reduced dataset of 607 hybridizations (Appendix A). Hybridizations were discarded according to different criteria:
−Replicates with large variation in expression values as detected by the Pearson correlation coefficients (calculated between replicate pairs). We removed replicates whose sample pairs coefficients were below the 0.90 threshold (Appendix A). The number of replicates removed for each sample is as follows: 3 for RT_Myc_3wks_infection, 3 for GiantCell, 3 for GallTissue_GiantCell, 2 for RT_LCM_arbuscular, 2 for RT_LCM_cortical, 2 for RT_LCM_adjacent, 1 for Nod_Naut1_SalsC, 1 for RT_CRR_72hpi, 1 for RT_CRR_96hpi, 1 for Root_A17_control. (19 columns in total).−Samples with extremely low (<1 × 10^6^) sums of the expression values. All replicates were removed (Appendix A). (30 columns).−Duplicated data. Replicates were removed at alternated lines (Appendix A). (24 columns).−Experiments with one measurement for each sample, so-called “single” replicate (Appendix A). (30 columns).


To understand how correlation analysis of gene function could be affected by the cleaning of the dataset, we compared Pearson correlation coefficients between *M. truncatula* genes across all hybridizations before and after the cleaning, both on the linear values and the Log-transformed ones. We first focused our attention on genes involved in the biosynthesis of saponins, creating a list of genes extracted from the literature and genes encoding enzymes that synthesize and transform isoprenoids in the cytosol (mevalonate pathway) [36] and in the plastid (non-mevalonate pathway, also known as the MEP/DOXP pathway) [37]. Saponins are a large class of secondary metabolites abundant in legumes, made of a carbohydrate attached to a terpenoid [38]. The flux to these compounds is large and several of the biosynthetic enzymes and precursors are known [39,40]. The saponins’ genes are listed in Appendix A. The putative mevalonate kinase (Mtr.41545.1.S1_at) was employed as a test gene (Appendix A) because we expected large changes in the correlation values for this gene against all the genes after the cleaning (see Figure 1). Indeed, the top correlators in the linear analysis (Appendix A, comparing Linear Original vs. Linear Cleaned; for the difference, see Appendix A) show relevant changes, and the cleaned dataset returns a list with much stronger consistency in gene function. This suggests, not surprisingly, that the mevalonate kinase is involved in isoprenoid and possibly saponins biosynthesis. For instance, the second best correlator, using the cleaned dataset, is MTR_1g017270 (squalene monooxygenase, Mtr.10468.1.S1_at), which shows a correlation value of 0.876, while, using the original dataset, the same pair of probes (Mtr.41545.1.S1_at vs. Mtr.10468.1.S1_at) gives a correlation of 0.597. Most of the genes at the top of the list behave in a similar way. This means that many of the expected correlators become concealed in the original dataset, and hence they are false negatives before cleaning. On the contrary, computing Spearman (Appendix A) or Kendall (Appendix A) correlation coefficients for the same probeset reveals that these methods identify top correlators with a strong biological consistency with both the original and the cleaned dataset. The biological consistency is comparable to Pearson’s Linear and Log analysis performed on the cleaned dataset. The two methods based on the rank correlation are, therefore, quite insensitive to the presence of strong outliers, as already known.

We also performed a search for genes targeted by different probes and identified a few more. We present five scatterplots (besides the usual Mevalonate kinase) in Appendix A, created with the whole dataset, not just the mean. As expected, cleaning tends to remove outliers and this results in an increase in the correlation value (B,C,F) in the linear scale, though much less dramatic than that for the mevalonate kinase (A), and a reduction in the log scale (D). One example (E) shows a large reduction in linear correlation.

Expanding the analysis to the saponins’ list (Appendix A), by comparing the linear correlation table before and after cleaning, it confirms the same trend, as evidenced in the “differential” table (Appendix A). Notably, the results of the Log analysis (Appendix A for the ‘One vs. All’ approach and Appendix A for the saponins’ gene list correlation table) returns a very different outcome. While most of the best correlators are still present in top positions in the Log analysis after cleaning, the Pearson coefficient shows an overall decrease across all the genes (Appendix A for the difference). The top correlators remain, therefore, quite consistent, at least in the case of Mtr.41545.1.S1. Again, the same holds true for the saponins’ genes correlation table (Appendix A). The same data are presented with two heatmaps (Figure 3) from which it is evident a reduction in the correlation values for many gene pairs, i.e., a reduction in false positives when passing from the uncleaned to the cleaned dataset. The correlation values were reduced on average by 0.26 (Appendix A), with many turning from a strong positive correlation (red/orange) to insignificant (white/light blue) (Figure 3). This means that there is in the original dataset something that increases most, if not all, correlation values in the Log analysis (see also the average differences in Appendix A), something that has no such an effect in the linear analysis. The cleaning, thus, shows a different, sometimes opposite, effect on the linear and the Log analysis. Again, focusing on the correlators of the mevalonate kinase (Mtr.41545.1.S1_at) with the saponins’ gene subset (original vs. cleaned dataset, Appendix A) as well as the differences between the values, we recognize the same pattern: correlation values in the linear analysis (Appendix A) increase on average, while those in the Log analysis (Appendix A) show a substantially larger decrease.

We observed comparable variations in correlation values for several families of transcriptional factors (TFs), one example being the bHLH (basic-Helix–Loop–Helix) family. bHLH is a large TF family, well characterized in eukaryotes and involved in several processes in plants, such as metabolism, growth and responses to stress [41,42]. Appendix A shows the list of *M.truncatula* bHLH genes selected from PlantRegMap (http://plantregmap.gao-lab.org/, accessed on 13 May 2021) [43,44]: in *M. truncatula,* this family included 168 genes in February 2020. An extensive decrease in correlation values is observable in the heatmaps generated from the logarithmic analysis of the bHLH family switching from the original to the cleaned datasets (Appendix A and Figure 4). In short, a great share of the actual correlation values in the Log analysis was reduced upon cleaning, suggesting that many top correlators in the original dataset are actually false positives. This appears to be a general phenomenon and the reason is the presence of a specific group of experiments. We provide an explanation for this phenomenon in the discussion. As performed for the saponins’ genes, we compared bHLH Log and linear Pearson correlation values (Appendix A) before and after the cleaning by means of ‘differential tables’ (Appendix A); we observed the same trend highlighted for the saponins’ dataset. Similar results were also observed for other TF families, such as WRKY, MYB, ERF and NAC (data not shown).

The bHLH gene family was also analyzed, using Spearman’s rank correlation on both the original and cleaned dataset (Appendix A), highlighting changes with a ‘differential table’ (sheet C, Cleaned–Original). Another differential table highlights differences between Pearson’s and Spearman’s coefficients (sheet D is Pearson’s linear coefficient, identical to Appendix A; sheet E, Pearson–Spearman). In this case, Spearman’s correlation seems to be quite insensitive to cleaning (average difference is 0.12, Appendix A), but there are instances of large differences between Pearson and Spearman’s coefficients calculated for the same pair of probes from the cleaned data. This means that the correlation measure employed may yield different results depending on the gene analyzed (Appendix A).

### 2.5. AgriGO

To further substantiate the effectiveness of the cleaning process, we also performed a GO enrichment analysis using AgriGO [45,46], working on co-expression results for selected genes using the SEA mining tool (see Discussion and Materials and Methods sections). We picked for each selected gene the 49 best correlators in the original and cleaned datasets. The AgriGO analysis was performed, from both linear and logarithmic correlation values, for the following genes as representatives of fundamental processes (glycolysis/respiration, translation, photosynthesis and gluconeogenesis): Mtr.31871.1.S1_at (pyruvate dehydrogenase E1 beta subunit, PDHE1-B, data not shown), Mtr.10637.1.S1_at and Mtr.34423.1.S1_at (60 ribosomal proteins, data not shown), Mtr.12203.1.S1_at (Rubisco small subunit, data not shown), Mtr.37533.1.S1_at (fructose 1,6-diphosphate phosphatase, data not shown) and Mtr.12230.1.S1_at (translation elongation factor EF-2 subunit, Figure 5 and Figure 6). Again, Mtr.41545.1.S1_at (the putative mevalonate kinase) was used as a test gene for which large changes were expected, which indeed was the case (data not shown). Correlators for Mtr_12330.1.S1_at from the original and cleaned datasets are shown in Appendix A. The variations in the enrichment of GO terms of co-expressed genes between the original and the cleaned dataset strongly suggest that, despite the small number of removed hybridizations, the cleaning improves the quality of the output for several genes and, therefore, its biological significance.

## 3. Discussion

Our analysis highlighted how the repository *Medicago truncatula Gene Expression Atlas* (*MtGEA*) [12,13], a reference database for microarrays studies of *Medicago* species, contains data that are difficult to explain due to biological or measurement variabilities.

The scatterplot presented in Figure 1 efficiently shows the co-expression of two probes for a putative mevalonate kinase (Mtr.41545.1.S1 and Mtr.16327.1.S1_at). It is possible to observe three points (indicated with red arrows) isolated from all the other ones. Since both probes refer to the same gene, a linear relationship between the two probes is to be expected. Being the only three outliers, we can imagine that they could be due to particular experimental conditions or tissues, or they could be due to errors in measurements or processing of the data. We identified the three outliers as corresponding to means of three samples (RT_LCM_arbuscular, RT_LCM_cortical and RT_LCM_adjacent, [31]), and we discovered that all of them were performed using the LCM technique, normally used to isolate small portions of tissue or just few cells [47]. With this method, an amplification step is usually required in order to have sufficient material for the hybridization. These outliers are therefore likely the result of spurious amplification and/or other technical problems in the experimental execution.

To understand if similar experiments could generate other outliers in scatter plots of other genes, we decided to use as a first criterion the sum of gene expression values of all probes to detect potential problems. For most hybridizations of the dataset, values of the sum were around 2 x 10^7^; they were, however, dramatically lower, even up to 100 times, in just a few cases (Appendix A). These problematic data are easily identifiable graphically, such as in Figure 2. The RT_LCM samples, however, not only show low values for the sum (Appendix A), but also a large difference among replicates for each of the three samples (Appendix A). Since replicates of the same tissue/treatment should give very similar expression values, the difference in replicates of the RT_LCM group confirms the possibility that they could be the result of systematic or methodological errors in the experimental procedure, compromising their validity.

We found six other groups of samples whose sum of expression values was lower than most of the samples of the dataset. Even though the reasons of the anomaly could be related to experimental problems for four groups (Appendix A), the groups in Appendix A have sums of expression values that are too low to be explained by experimental variability. The first group is associated to one study [32] and includes five samples, each one with three replicates of specific root cell types *Medicago truncatula* in relationship with *Glomus intraradices* mycorrhizal fungus. APP_P and NAP_C are samples at 5–6 days post inoculation (dpi), while ARB-A, CMR_K and EPI_E refer to 21 dpi. The second group includes five samples, each one with three replicates [33]. In this second study, samples from the meristem were analyzed, both in the region infected by the bacterium *Sinorhizobium melitoti* and in the proximal region. These two groups show systematic errors possibly introduced during the insertion of data in MtGEA, or already present in the studies. We analyzed the corresponding values in the downloaded dataset from MtGEA (Appendix A). Expression values for these experiments are too low to be biologically relevant, especially for the so-called housekeeping genes, which tend to have a fairly constant expression in organisms. Moreover, the mean values of the replicates calculated for these experiments from the original data were different from the downloaded mean values in Appendix A (data not shown). The most likely error is the insertion of the data for these two experiments after transformation into logarithmic values, even though the means were calculated based on original values (not Log-transformed). This was confirmed by performing an antilog transformation of the values and then measuring their mean values: the results were identical or nearly identical to the mean values uploaded in MtGEA (data not shown).

To point out other anomalies in the experiments, we decided to use the Pearson correlation analysis among sample replicate pairs as the second criterion for the cleaning of the dataset. Using the Pearson correlation coefficient, we could compare the experiments to one another across around 50,000 genes. Thanks to this second criterion, we confirmed that not only the previously identified samples were problematic, but that also other groups of samples could give significant anomalies. Pearson correlation coefficients were analyzed only for samples with at least two replicates: samples with a single hybridization (Appendix A) were not considered since we could not test them according to this criterion and were, in any case, few. One study argues for their removal [6]. We considered threshold values for correlation coefficients as 0.90 and, according to the results of the analysis, four possible cases could occur: (1) values of the correlation coefficients for all replicates studied are above the threshold value, thus all data are retained; (2) one correlation coefficient for three replicates is below the threshold value, thus only the pair that has the highest value above the threshold is kept; (3) two correlation coefficients are below the threshold value, thus only the pair showing a correlation value above the threshold is kept; and (4) all correlation values measured are below the threshold value, thus all replicates are discarded. By excluding samples with one single hybridization (Appendix A), we automatically discarded also those in Appendix A for which we established a possible systematic error in the data. Performing the analysis on all the other samples, we highlighted correlation values below the threshold of 0.90 as presented in Appendix A. Moreover, analyzing these results, we highlighted that the last two groups of Appendix A have identical correlation values. Checking the original data (Appendix A), we discovered that there are 24 columns duplicated (8 samples each with 3 replicates), due to an error during data insertion in MtGEA. One first group refers to a study [35] on nodules treated with phosphinothricin at different times. A second study presents data on the roots of different genotypes in relationship to different symbionts [34]. We also identified another group of experiment whose names were duplicated [12]. This groups caused problems in correctly reading the original dataset since not all duplicated columns were filled, generating problems in the organization of data in columns and in the analysis.

From our work on the MTGEA database (https://mtgea.noble.org/v3/, accessed on 13 May 2021), we discovered that the repository contains errors and poor quality data that affect subsequent analyses. After the cleaning of MtGEA, using the criteria of the sum of the expression values and the Pearson correlation coefficients among replicates pairs, we reduced the number of columns in the dataset from 716 (710 once removed the duplicated names of two samples in the dataset heading—Nod_10dpi and Nod_14dpi [12]) to 607, actually removing 103 hybridizations (around 15%). To verify that the cleaning results in statistically significant improvements, we performed Pearson correlation analyses among genes of different pathways/processes out of the original and cleaned datasets. In order to reduce the number of false positive correlations with no biological significance, it is important to remove unreliable and misleading data. We first studied the saponins biosynthetic pathway, which is particularly important in *Medicago truncatula*, as well as in other leguminous plants used for animal feeding. Saponins, because of their bitter taste, can reduce the appeal of the feed and can be toxic [48]. *Medicago* spp. saponins are triterpenic saponins derived from 2,3-oxidosqualene, the last common precursor of sterols and triterpenes, synthesized into the cytosol from isopentenyl pyrophosphate (IPP) [39,49]. The positive effect of MtGEA cleaning on Pearson correlation analysis can be immediately observed in Figure 3. Comparing the heatmaps obtained from the Log-trasformed original and cleaned data for the genes involved in saponins biosynthesis (Appendix A), it is evident a drastic reduction in correlation strength of many genes, meaning that the cleaning reduced the number of false positives.

We also compared the results obtained on some genes with Pearson to those obtained with rank correlation methods. Spearman and Kendall’s correlation performed well and were not much affected by the cleaning; however, the results did not always coincide with Pearson’s correlation (Appendix A). This is a rather trivial observation because Pearson’s measure detects linear correlation, while rank correlation methods detect also other forms of correlation (e.g., hyperbolic, sigmoid…). For this reason, it may be interesting to use both Pearson’s (both Lin and Log) as well as Spearman’s for correlation analysis when searching for candidate genes. Kendall’s is more computationally intensive, and it could be used only for specific cases.

For some specific genes, the results of the correlation analysis after the cleaning change significantly, also ameliorating the results of the co-expression analysis. This is easily observable analyzing Appendix A, showing the correlators of the putative mevalonate kinase Mtr.41545.1.S1_at. Before the cleaning, this gene showed just a few high correlations with genes in the list of saponins biosynthetic genes, while after the cleaning, its biological consistency increases significantly. With the original data, Mtr.41545.1.S1_at best correlators are mostly genes without annotation or involved in pathways unrelated to saponins biosynthesis, such as ribosomal proteins. After cleaning, Mtr.41545.1.S1_at correlates with many genes involved in isoprene and saponins biosynthesis, meaning that the cleaning improved the biological significance of the correlation. In fact, 11 of the 15 best Log correlators of Mtr.41545.1.S1_at, using cleaned data, are attributable to isoprenoid biosynthesis (IPP), particularly in the mevalonate cytosolic biosynthetic pathway (Appendix A) and only four with the original data. This underlines that the cleaning not only reduced the number of false positives, but also reduced the number of false negatives (genes that were not well correlated before the cleaning but improved thereafter).

The importance of the cleaning of MtGEA before running a correlation analysis is also evident upon analyzing the Pearson correlation coefficients for several TFs family, for instance, the bHLH gene family (Figure 4 and Appendix A). Again, a great reduction in false positives is evident when switching from the original to the cleaned dataset. The cleaning helps in ameliorating the results of the correlation analysis, reducing the number of spurious correlations and helping in focusing only on strong correlations that can have a biological significance. Analogous results have been found for other TFs, such as WRKY, MYB, ERF and NAC (data not shown), leading us to believe that false positives are mainly found in the Log analysis when working on original data of TFs. In some cases, specific predictions can be made on the basis of the best correlators; for instance, the probe Mtr.34810.1.S1_at (Mtr_8g065740) refers to a TF, without further annotation. On the basis of the correlators, we anticipate for this gene a role in chromosome maintenance/stability/DNA repair because half of the best 20 correlators in the Log analysis fall into this category (data not shown). This is not at all evident in the list obtained with the original data. The same is true for the linear analysis, albeit the numbers are less striking.

As another example, probe Mtr.5966.1.S1_at, which identify a class III peroxidase, correlates best with two probes (Mtr.42141.1.S1_s_at and Mtr.42141.1.S1_at) in the linear analysis; both probes recognize another peroxidase (MtPRX1, MTR_3g094630) [50]. The very same probes are at positions 9 and 10, respectively, in the correlators list generated with the original dataset, implying a significant change in their degree of correlation with probe Mtr.5966.1.S1_at (data not shown). All three probes refer to transcripts strongly induced by elicitor treatments [51]. Surprisingly, the correlation analysis for another peroxidase (PRX3, Mtr.40125.1.S1_at), apparently not involved in aurone biosynthesis [50], suggests a strong involvement in disease resistance, a conclusion based on the frequency of GO terms among the best correlators. This is evident mainly in the Log analysis (both of original and cleaned datasets) and suggests that it is always worth performing a Log analysis beside the linear one.

AgriGO analysis, on original and cleaned datasets, was also used to measure the improvement in biological consistency. AgriGO is a web tool that allows to perform the gene ontology analysis, focusing the attention on species of agricultural interest [45,46]. Different mining tools are available; we used the Singular Enrichment Analysis (SEA) that provides an enrichment analysis of GO terms for a list of genes/probes, with the aim to find GO terms that are statistically enriched in a list compared to an expected value for a given species. This analysis was performed for several genes, using the respective 49 best correlators. Several of the analyzed genes show improvements, for example: Mtr.41545.1.S1_at (putative mevalonate kinase, data not shown), Mtr.31871.1.S1_at (pyruvate dehydrogenase E1 beta subunit, PDHE1-B, data not shown) and Mtr.12230.1.S1_at (translation elongation factor EF-2 subunit, Figure 5 and Figure 6 and Appendix A). The AgriGO analysis for EF-2 highlights a great change in the enriched GO terms before and after the cleaning, evidenced by an intensification of the red color, both in linear and logarithmic forms, after the cleaning. Comparing original and cleaned data, the identified processes are the same, but there is a great difference in significance levels, which increase upon cleaning.

It is noteworthy that our approach is not only conceptually and computationally very simple, but it does not require prior knowledge of the biological samples or the species, differently to methods for the removal of unwanted variations that are based on normalizations with respect to a set of control genes [52,53,54]. These other methods can, thus, be seen as complementary and could be applied before or after processing the datasets according to our approach.

## 4. Conclusions

Our analysis on the MtGEA database has not only improved the quality of the data but underlined the importance of a proper cleaning step before any kind of correlation analysis on microarray data. Moreover, we have established a simple strategy of general validity for the cleaning of microarray datasets based on two criteria: the sum of the expression values across all genes in samples and the Pearson correlation analysis among sample replicate pairs. We demonstrated how the cleaning can strongly affect the transcript correlation analysis. We found that the removal of a limited number of problematic samples ameliorates the results of the correlation analysis (both reducing false positives and false negatives) and, consequently, of related predictions that are more supported by the annotation, literature and GO terms’ frequency. We believe that this approach is of general applicability and could be expanded beyond the *Affymetrix* technology.

## 5. Materials and Methods

### 5.1. Microarray Data

The microarray data used in this study were downloaded from the *Medicago truncatula Gene Expression Atlas* (MtGEA) (https://mtgea.noble.org/v3/, accessed on 13 May 2021) [12,13]. The dataset consists of gene expression profiles from 36 experiments. It was generated by selecting as download options “All Replicates” for ”Experiment Selection” and “Mtr:Medicago truncatula only” for “Probeset Selection”. As of February 2020, the complete dataset was composed by 50,900 genes with 710 hybridizations (the downloaded dataset initially included 716 columns because of the name duplication of two samples’ replicates, see Results and Discussion sections for further details). The dataset, including the mean values of experimental replicates (274 columns), was downloaded by selecting “All Means” and “Mtr: Medicago truncatula only”.

### 5.2. R

Data analysis was conducted in the R programming environment (https://www.R-project.org/, version 4.0.0) [55], and figures were produced using the following packages: *ggplot2* (https://cran.r-project.org/web/packages/ggplot2/index.html) [56], *pheatmap* (https://cran.r-project.org/web/packages/pheatmap/index.html) [57], *data.table*
https://cran.r-project.org/web/packages/data.table/index.html [58], *Hmisc* (https://cran.r-project.org/web/packages/Hmisc/index.html) [59] and *RColorBrewer* (https://cran.r-project.org/web/packages/RColorBrewer/index.html) [60]. Annotations were performed and checked using g:Profiler (https://biit.cs.ut.ee/gprofiler/gost) [61] and the *Affymetrix* microarray annotations for *Medicago* (http://tools.thermofisher.com/content/sfs/supportFiles/Medicago-na36-annot-csv.zip). All pages were accessed on 15 April 2021.

### 5.3. AgriGO

Gene ontology analysis was performed using AgriGO (http://bioinfo.cau.edu.cn/agriGO/index.php, accessed on 14 May 2021) [45,46], while Singular Enrichment Analysis (SEA) was the tool used to identify GO terms statistically enriched in a provided list of genes.

## Figures and Tables

**Figure 1 plants-10-01240-f001:**
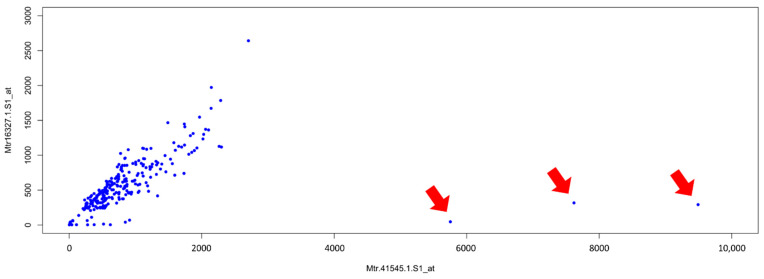
Scatterplot of the mean expression levels of probes Mtr.41545.1.S1_at and Mtr.16327.1.S1_at. Three outliers are indicated by red arrows.

**Figure 2 plants-10-01240-f002:**
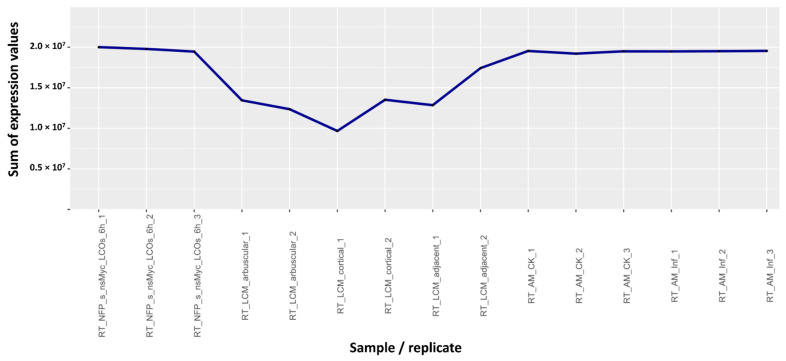
Graphical representation of the sum of the expression values for the RT_LCM samples [31], compared to the ones adjacent to them in the dataset list. All the other samples have sum of expression values close to 2.0 × 10^7^, whereas the RT_LCM samples range between 1.0 × 10^7^ and 1.7 × 10^7^.

**Figure 3 plants-10-01240-f003:**
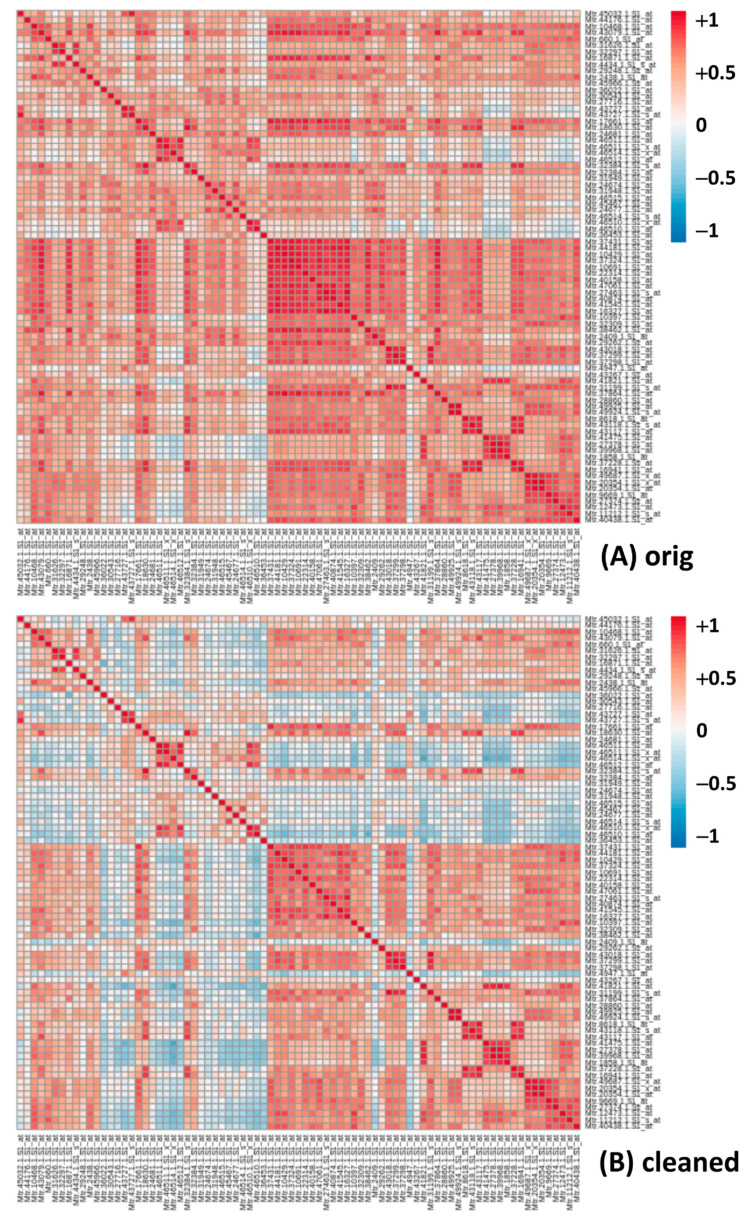
Heatmaps for Pearson correlation coefficients in the logarithmic form for genes of the saponins biosynthetic pathway, with expression values from original (**A**) and cleaned (**B**) Log datasets (Appendix A).

**Figure 4 plants-10-01240-f004:**
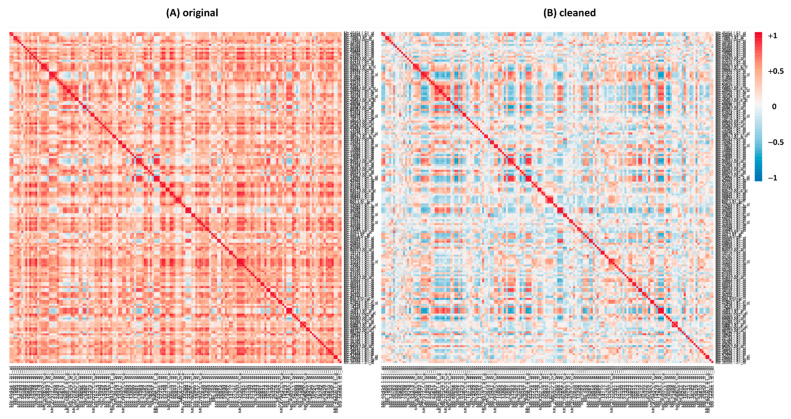
Heatmaps generated for Pearson correlation coefficients in the logarithmic version for the TFs family bHLH, with expression values from original (**A**) and cleaned (**B**) Log datasets (Appendix A).

**Figure 5 plants-10-01240-f005:**
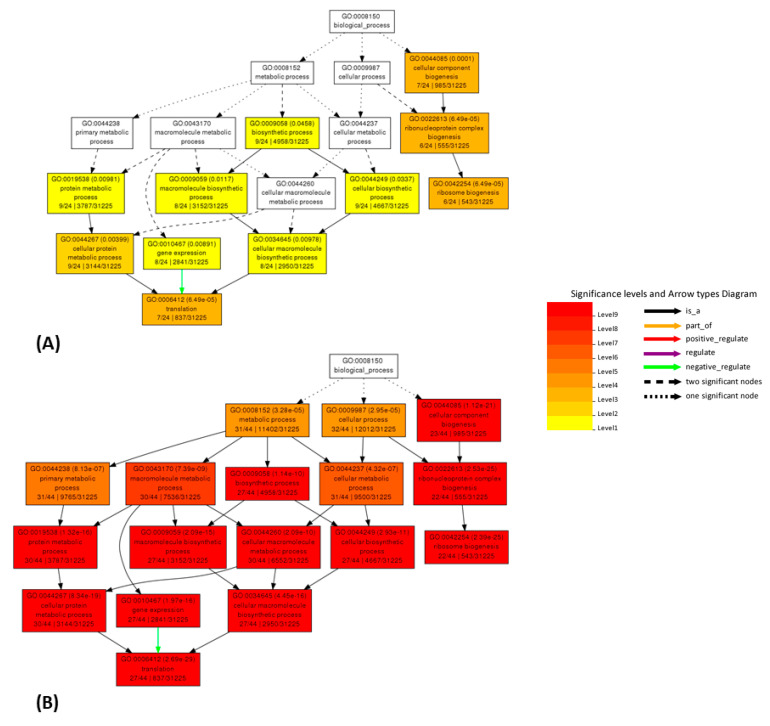
Comparison of AgriGO enrichment analysis working with linear co-expression data for the Mtr.12230.1.S1_at gene (Appendix A). Results obtained from correlation values working on the original (**A**) and cleaned (**B**) dataset.

**Figure 6 plants-10-01240-f006:**
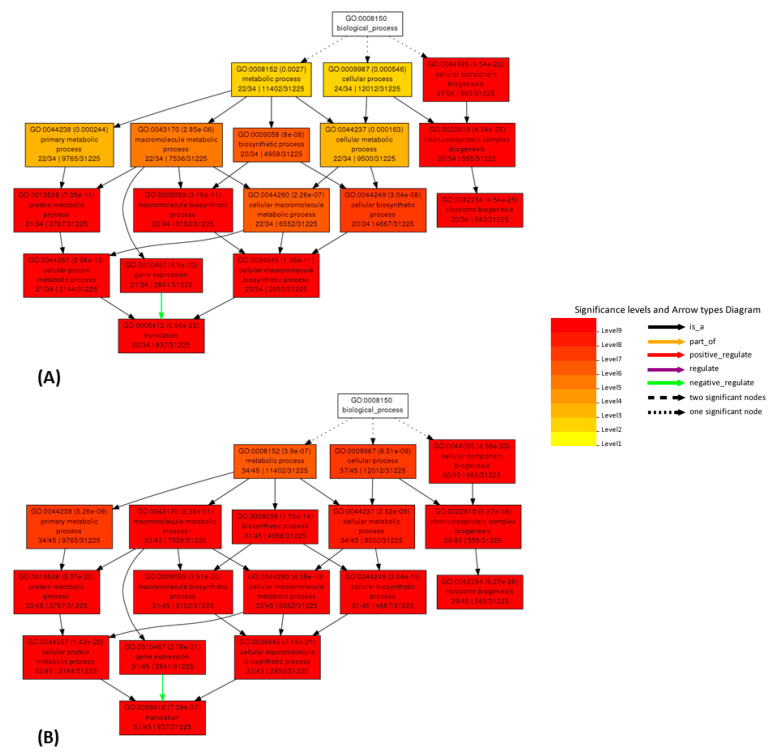
Comparison of AgriGO enrichment analysis working with logarithmic co-expression data for the Mtr.12230.1.S1_at gene (Appendix A). Results obtained from correlation values from the original (**A**) and cleaned (**B**) dataset.

## Data Availability

The data presented in this study are either openly accessible at the original repositories (as detailed in Material and Methods) or provided as Appendix A.

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
