# Peer review of "Cleaning the Medicago Microarray Database to Improve Gene Function Analysis"

_plants, 2021, doi:10.3390/plants10061240_

Round 1
Reviewer 1 Report
The authors described a protocol for cleaning the Medicago microarray data MtGEA through 3 steps :
- Sample cleaning by comparing sums of the expression levels and removing duplicates
- Replicate cleaning by evaluating correlations between replicates from a same sample (removing replicates with correlation below 0.9)
- Removal of samples with a single replicate
The resulting cleaned dataset showed a reduction in both false positive and false negative as suggested by improved GO analysis tested on a subset of genes known to be involved in saponin biosynthesis. The Pearson correlations between original and cleaned dataset (Log transformed or not) for this subset of genes were compared and discussed. Additional comparison was made for the bHLH transcription factor family.
Main concerns :
- A cleaning procedure to remove unwanted variation in large microarray gene expression datasets and allow the estimation of the true underlying gene-gene correlations has already been published : Freytag, S., Gagnon-Bartsch, J., Speed, T.P. et al. Systematic noise degrades gene co-expression signals but can be corrected. BMC Bioinformatics 16, 309 (2015). https://doi.org/10.1186/s12859-015-0745-3. A R package for correcting that noise is available at http://www.bioconductor.org/packages/release/bioc/html/RUVcorr.html. It would be interesting for the authors to test this RUVcorr program and compare their results in terms of cleaning efficiency and candidate gene prioritization.
- Study of mean expression values between samples and global correlations between replicates are essential steps in cleaning. However, a comprehensive study of discordant expression observed for different probesets that match the same gene would be interesting for the Medicago scientific community. Only one example (Figure1) is included in the present manuscript.
- The comparison of Pearson correlation for raw data or log-transformed data is not convincing. The Pearson correlation measure the linear dependence between two vectors. As log-transformation is a non linear transformation, the correlations are expected to change. If the authors really want to compare correlations between these two datasets (raw and log-transformed), I suggest to perform rank correlation instead of Pearson (Kendall or Spearman correlations).
Minor concerns :
Some sentences were not clear for me:
89-90 (“change in gene function prediction for a number of genes” ?) -> change in GO categories overrepresented in the geneset ?
370-372. Please rewrite.
Reviewer 2 Report
This paper by Marzorati et al. showed to have an experimental results focusing on the quality improvement of microarray databases. Data quality improvement was confirmed through correlation analysis after data correlation analysis and data cleaning for the MtGEA database, and related data were provided as supplementary figures and tables. According to the Pearson correlation analysis, the quality improvement before and after data cleaning was found to have increased.
However, since it was described that the purpose of this study is to improve grass quality through general methods based on logical and statistical relationships in the use of R programs in materials and methods, it would be needed to provide R scripts used in this study.
It is detected over 11% similar text in this article. So authors need to revise some parts.
Round 2
Reviewer 1 Report
Authors have answered all my queries and made substantial changes and additions to the manuscript.